# Wax worm saliva and the enzymes therein are the key to polyethylene degradation by *Galleria mellonella*

A. Sanluis-Verdes[1,9], P. Colomer-Vidal [1,9], F. Rodriguez-Ventura[1], M. Bello-Villarino [1], M. Spinola-Amilibia[2], E. Ruiz-Lopez[3], R. Illanes-Vicioso [3], P. Castroviejo [4], R. Aiese Cigliano[5], M. Montoya [6], P. Falabella [7], C. Pesquera[8], L. Gonzalez-Legarreta[8], E. Arias-Palomo [2], M. Solà[3], T. Torroba[4], C. F. Arias [1] ✉ & F. Bertocchini [1] ✉

Plastic degradation by biological systems with re-utilization of the by-products could be a future solution to the global threat of plastic waste accumulation. Here, we report that the saliva of *Galleria mellonella* larvae (wax worms) is capable of oxidizing and depolymerizing polyethylene (PE), one of the most produced and sturdy polyolefin-derived plastics. This effect is achieved after a few hours' exposure at room temperature under physiological conditions (neutral pH). The wax worm saliva can overcome the bottleneck step in PE biodegradation, namely the initial oxidation step. Within the saliva, we identify two enzymes, belonging to the phenol oxidase family, that can reproduce the same effect. To the best of our knowledge, these enzymes are the first animal enzymes with this capability, opening the way to potential solutions for plastic waste management through bio-recycling/up-cycling.

Polyethylene (PE) accounts for 30% of synthetic plastic production, largely contributing to plastic waste pollution on the planet to-date[1]. Together with polypropylene (PP), polystyrene (PS) and polyvinylchloride (PVC), PE is one of the most resistant polymers, with very long C–C chains organized in a crystalline, dense structure. Given the hundreds of million tons of plastic waste accumulating and the still escalating pace of plastic production, re-utilization of plastic residues is a necessary path to alleviate the gravity of the plastic pollution problem, and at the same time to render available a huge potential reservoir of carbon[2]. To-date, only mechanical recycling is being applied at a large scale. Several factors, such as the low number of plastic types prone to be mechanically recycled, and the low quality of the secondary products severely restrict the potential of this solution to the problem of plastic waste accumulation. Chemical recycling, as

an alternative procedure, is preferentially aiming at plastic upcycling, e.g., decomposing polyolefin-derived plastics in order to take advantage of smaller intermediates. Several technologies have been applied at a lab scale, although the high energetic cost might still impede the scaling up of these technological tools[3].

In addition to mechanical and chemical recycling, biodegradation is widely considered as a promising strategy to dispose of plastic residues. Biodegradation refers to environmental degradation by biological agents. The IUPAC defines biodegradation as the "breakdown of a substance catalyzed by enzymes in vitro or in vivo"[4], later modified "to exclude abiotic enzymatic processes"[5]. In the case of PE, biodegradation requires the introduction of oxygen into the polymeric chain[6,7]; this causes the formation of carbonyl groups and the subsequent scission of the long hydrocarbon chains with the

[1]Centro de Investigaciones Biologicas-Margarita Salas (CIB)-Consejo Superior de Investigaciones Cientificas (CSIC), Department of Plant and Microbial Biology, Madrid, Spain. [2]CIB-CSIC, Department of Structural and Chemical Biology, Madrid, Spain. [3]Department of Structural Biology, Molecular Biology Institute of Barcelona (IBMB)-CSIC, Barcelona, Spain. [4]Department of Chemistry, Faculty of Science and PCT, University of Burgos, Burgos, Spain. [5]Sequentia Biotech SL, Barcelona, Spain. [6]CIB-CSIC, Department of Molecular Biomedicine, Madrid, Spain. [7]Department of Sciences, University of Basilicata, Potenza, Italy. [8]Department of Chemistry and Process & Resource Engineering, Inorganic Chemistry Group-University of Cantabria, Nanomedicine-IDIVAL, Santander, Spain. [9]These authors contributed equally: A. Sanluis-Verdes, P. Colomer-Vidal. ✉e-mail: tifar@ucm.es; federica.bertocchini@csic.es

production of smaller molecules, which can then be metabolized by microorganisms[8,9]. The crucial first step of this chain of events, i.e., the oxidation of PE polymer, is usually carried out by abiotic factors such as light or temperature[6,8,10]. Once the long polymeric molecules are broken down, a process that takes years of exposure to environmental factors in the wild, bacteria or fungi intervene and continue the job[6,9,11,12]. This is the current paradigm driving the research field in biodegradation. Within this paradigm, several bacterial and fungal strains have been identified as capable of carrying on a certain extent of PE degradation. However, in most of the cases, such degradation requires an aggressive pre-treatment of PE (heating, UV light, etc.) that accelerates the incorporation of oxygen into the polymer, making the abiotic oxidation the real bottleneck of the reaction[12–16]. In the past decade, a few microorganisms have been described as capable of acting on untreated PE[17–23], although they require a significantly longer incubation time compared to experimental conditions with pre-oxidized PE.

The identification of enzymes from microorganisms capable of degrading untreated PE has proven a difficult task. In fact, to our knowledge, no such enzyme has been identified. Reported enzymes capable of acting on polyolefin-derived plastics require a pretreatment of the plastic material[12,14]. For example, two reported laccases, able to chemically modify PE, necessitate an abiotic pretreatment[24] or the addition of redox mediators such as 1-hydroxybenzotriazole[25]. Abiotic pre-treatments such as radiation or heat cause oxidation of the polymer, which is the crucial limiting step in the biodegradation chain[12].

This scenario confirms that the synthetic nature of the compound, together with the hydrophobicity and inaccessibility features, make plastic a difficult target for animal, fungal or microbial-derived enzymatic activities. Nonetheless, some lepidopteran and coleopteran insects revealed the capacity to degrade untreated PE and PS[26–31]. The larvae of *Galleria mellonella*, also known as wax worms (ww), can oxidize PE within one hour from exposure[29,32–38], making it the fastest known biological agent capable of chemically modifying PE.

Our work here begins with an initial observation that plastic debris appears when a PE film is in contact with the recently formed ww cocoon and mouth secretions. We then perform a detailed analysis of the saliva of the ww (GmSal), revealing the capacity of GmSal to oxidize and break PE within a time frame of a few hours. This effect is confirmed by Gel Permeation Chromatography (GPC) analysis of GmSal-treated PE, showing the scission of long hydrocarbon chains into small molecules. Using Gas Chromatography-Mass Spectrometry (GC-MS) we identify degradation products such as small oxidized aliphatic chains, further confirming the breaking of the polymer into shorter molecules. Proteomic analyses of GmSal reveals the presence of a handful of enzymes belonging to the hexamerin/prophenoloxidase family. Two of these enzymes are shown to oxidize PE after a few hours' application at room temperature (RT). For this reason, we named them PEases. These PEases are an arylphorin here re-named Demetra, and an hexamerin here re-named Ceres. In the case of Demetra, we show that PE oxidation is accompanied by the deterioration of PE and the release of degradation by-products similar to those obtained with the whole saliva.

To the best of our knowledge, these PEases are the first enzymes capable of producing such modifications on a PE film working at room temperature and in a very short time, embodying a promising alternative to the abiotic oxidation of plastic, the first and most difficult step in the degradation process. The identification of invertebrate enzymes capable of oxidizing PE in a few hours represents an alternative paradigm in the world of plastic degradation and more widely in the plastic waste management fields, opening up various possibilities which may help to solve the plastic waste pollution issue, along with expanding potential formulae/routes for synthetic polymer production.

## Results

### Wax worm saliva oxidizes PE film

Saliva, broadly defined here as the juice present in the anterior portion of the digestive apparatus, was collected from the ww mouth and tested on a commercial PE film (Fig. 1A). After three consecutive applications of 30 µl of GmSal for 90 min each, Confocal Raman microscopy/Raman spectroscopy (RAMAN) analysis indicated polymer oxidation, accompanied by a general deterioration of the film (Fig. 1B). This is evident in the overlapping with the PE control (Fig. 1C, D), which reveals the expected PE signature profile. As a further control, the saliva of another lepidopteran larva, *Samia cynthia*, was applied on the PE film, and no oxidation was generated (Fig. 1E). The changes produced by the GmSal in a few hours-long applications are similar to those generated by environmental factors after months or years of exposure to weathering[39,40]. The Fourier Transformed Infrared Spectroscopy (FTIR) analysis confirmed the oxidation profile (Fig. S1). The changes in PE chemical composition revealed by the spectroscopy techniques suggested that molecules other than the long PE polymeric chain formed upon the contact with GmSal.

### Wax worm saliva degrades PE

The molecular weight characteristics of PE before and after treatment with GmSal were analyzed using High Temperature-Gel Permeation Chromatography (HT-GPC). After a few hours' applications of GmSal on PE film (15 applications, 90 min each), the molecular weight distribution became bi-modal, with a new peak in the low molecular weight region, indicating the breaking of the C–C bonds with the appearance of small compounds (Fig. 2A). Formation of C=O bonds as a consequence of chain scission was shown by the increase of the carbonyl index (CI, Fig. 2A). The weight average ($Mw$) was only slightly changed (from 207,100 to 199,500 g/mol) suggesting that the polymer was not uniformly modified by the GmSal, with portions that have been strongly depolymerized, while others remained still untouched[41]. The same result was obtained using PE 4000 instead of PE film, with a change in $Mw$ from 4000 to 3900 g/mol and an increase in the carbonyl index in the GmSal-treated sample (Fig. 2B, black and blue curves). These results show that PE was oxidized and depolymerized as a consequence of GmSal exposure, with the formation of oxidized molecules of low molecular weight. In order to analyze the dynamics of changes in the PE after increased exposure to GmSal in time, we doubled the exposure time of PE 4000 to GmSal (30 applications, 90 min each) (Fig. 2B, red curve). Breaking of large polymer chains with the formation of smaller molecules notably increased at longer exposure times, as showed by the comparison of the molecular weight distributions and by the doubling of the CI (Fig. 2B). These experiments confirmed the PE oxidation with depolymerization upon a few hours' exposure to GmSal, an effect that increased with time, and pointed to the presence therein of still unknown activities capable of PE degradation.

### Identification and analysis of the by-products of PE treated with wax worm saliva

To have an insight into the degradation products resulting from PE-saliva contact and released from the oxidized polymer, PE granules (crystal polyethylene-PE 4000) were exposed to GmSal and subsequently analyzed by Gas Chromatography-Mass Spectrometry (GC-MS) and identified by NIST11 library for untargeted compounds. After 9 applications of 40 µL of GmSal for 90 min each at RT, new compounds were detected in the experimental sample (Fig. 3). The detected compounds comprised oxidized aliphatic chains, like 2-ketones from 10 to 22 carbons. Ketones from 10 to 18 carbons were identified by comparing the fragmentgram of the ion m/z 58 from methyl ketones that correspond with the transposition of the McLafferty on the carbonyl located at the second carbon of each ketone. Those not present in the library as 2-eicosanone and

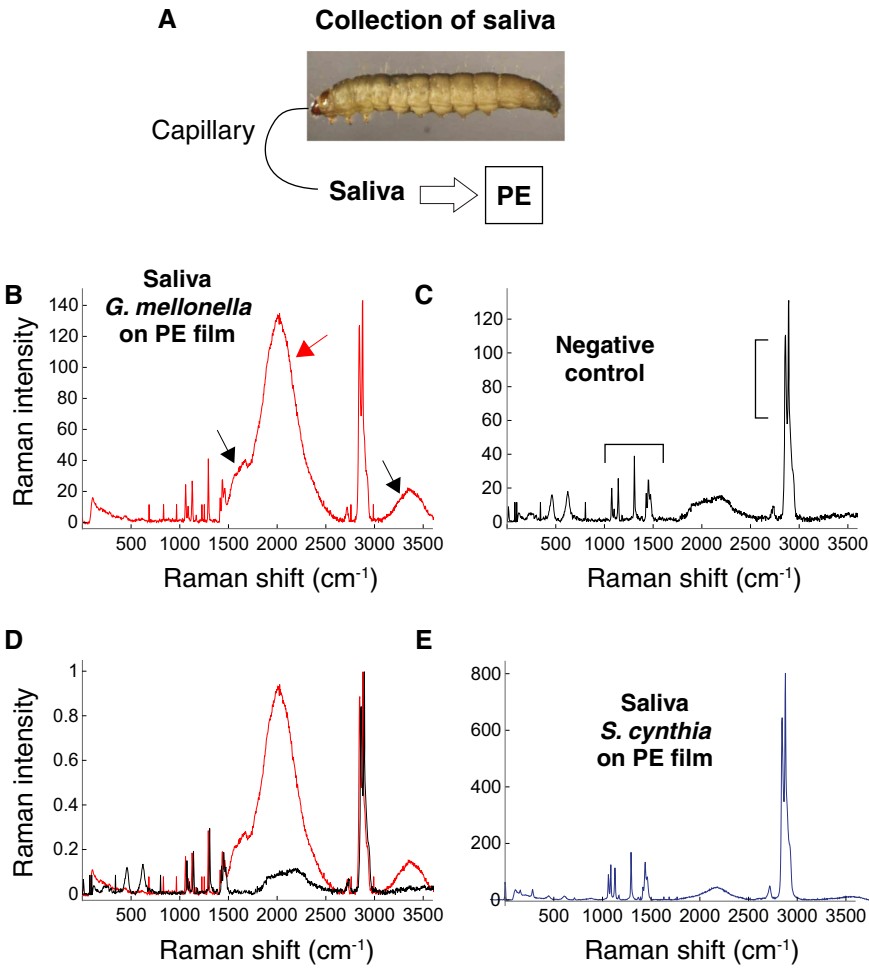

**Fig. 1 | *Galleria mellonella* saliva (GmSal) collection and functional study.**
**A** Scheme of saliva collection and application. **B**–**E** RAMAN analysis of PE film. **B** PE film treated with GmSal: 3 applications of 90 min, 30 µl each. The peaks between 1500 and 2400 cm⁻¹ indicate different collective stretching vibrations due to the presence of other organic compounds, sign of PE deterioration (red arrow). Oxidation is indicated between 1600 and 1800 cm⁻¹ (carbonyl group) and

3000–3500 cm⁻¹ (hydroxyl group) (black arrows)[78]. **C** Control PE film. Brackets indicate the peaks that characterize PE (PE signature), corresponding to the bands at 1061, 1128, 1294, 1440, 2846, and 2880 cm⁻¹. **D** Overlapping profiles (**B** and **C**). **E** PE film treated with *Samia cynthia* saliva. Source data are provided as a Source Data file.

2-docosanone showed the same fragmentgram m/z 58 and were defined by the equidistance of the peaks along the retention time and their molecular weight (Supplementary Table 1). Furthermore, the presence of 2-ketones was confirmed with GC-MS/MS in MRM mode with the ion with the highest m/z and the exclusive of each molecule as 212 > 58 for 2-tetradecanone, 240 > 59 for 2-hexadecanone, and 282 > 58 for 2-octadecanone. Also, butane, 2,3-Butanediol, 2-trimethylslyl (TMS) derivative, and sebacic acid, 2TMS derivative were identified using sample silylation, indicating the deterioration of the PE chain (Fig. 3C). At the same time, a small aromatic compound recognizable as benzenepropanoic acid, TMS derivative, a plastic antioxidant, was found. Derivative chemicals were confirmed as well using GC-MS/MS with an m/z of 147 > 73, 331 > 73, and 104 > 75, respectively. The presence of this plastic antioxidant suggests an "opening" in the polymeric structures, with the release of small stabilizing compounds normally present in plastics (plastic additives).

To verify if an increase in time exposure to GmSal caused an increase in PE degradation, we repeated the experiment of PE 4000 exposure in sequential times, with four applications per day of 100 µL of GmSal for 90 min each at RT, performed in 1, 2, 3 and 6 consecutive days. The analysis of the supernatants revealed a progressive increase in the formation of 2-decanone, 2-dodecanone, 2-tetradecanone, and

2-hexadecanone (Fig. S2). These data indicate an increase of at least twice the relative abundance of degradation products with time (from 1 to 6 days) as a consequence of prolonged exposure to the GmSal.

## Study of the wax worn saliva: enzymes identification and functional studies

To understand the nature of the buccal juice, a GmSal sample was analyzed by negative staining electron microscopy (EM), revealing a high content of proteins or protein complexes (size: 10–15 nm) (Fig. 4A). No other structures (such as vesicles or bacteria) were detected. Electrophoretic analysis (SDS-PAGE) confirmed the presence of proteins with a prominent band at around 75 kDa (Fig. 4B).

Does GmSal contain enzymes responsible for the detected PE modifications? To assess this, a proteomic analysis of GmSal contents was carried out. More than 200 proteins were detected, including a variety of enzymatic activities, transport and structural proteins, etc. (Supplementary Data 1). To narrow down the number of potential candidates, a saliva sample was analyzed by size exclusion chromatography (SEC). The elution profile showed a main single, wide peak (Supplementary Fig. 3A). SDS-PAGE gel of the major fraction showed a strong band at about 75 kDa (Supplementary Fig. 3C). The proteomic profile of this band revealed the presence of proteins known in arthropods as related to transport or storage.

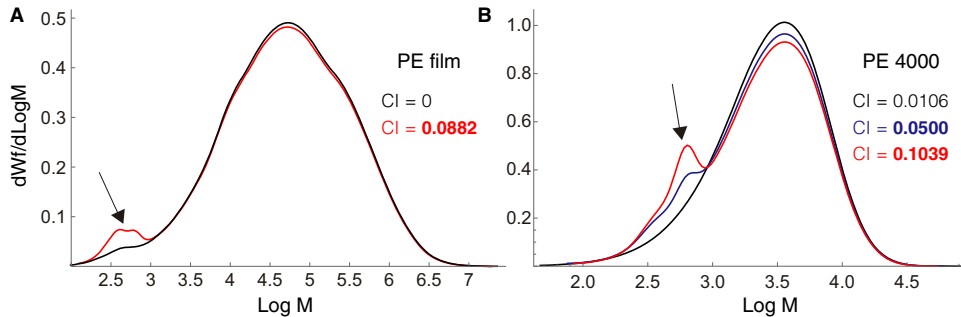

**Fig. 2 | HT-GPC analyses of PE treated with GmSal.** Molecular weight distribution of PE film (**A**) and PE 4000 (**B**) are indicated. **A** Control PE (in black) and PE-treated (in red) are compared. The carbonyl index (CI) of the control and experimental are indicated. $M_w$ control (red): 0; $M_w$ experimental (black): 0.0882. **B** Control PE (black), treated-PE (15 applications) (blue), treated-PE (30 applications) are compared. The CI of each sample is indicated. Arrows indicate compounds with low molecular weight in the treated PE. Source data are provided as a Source Data file.

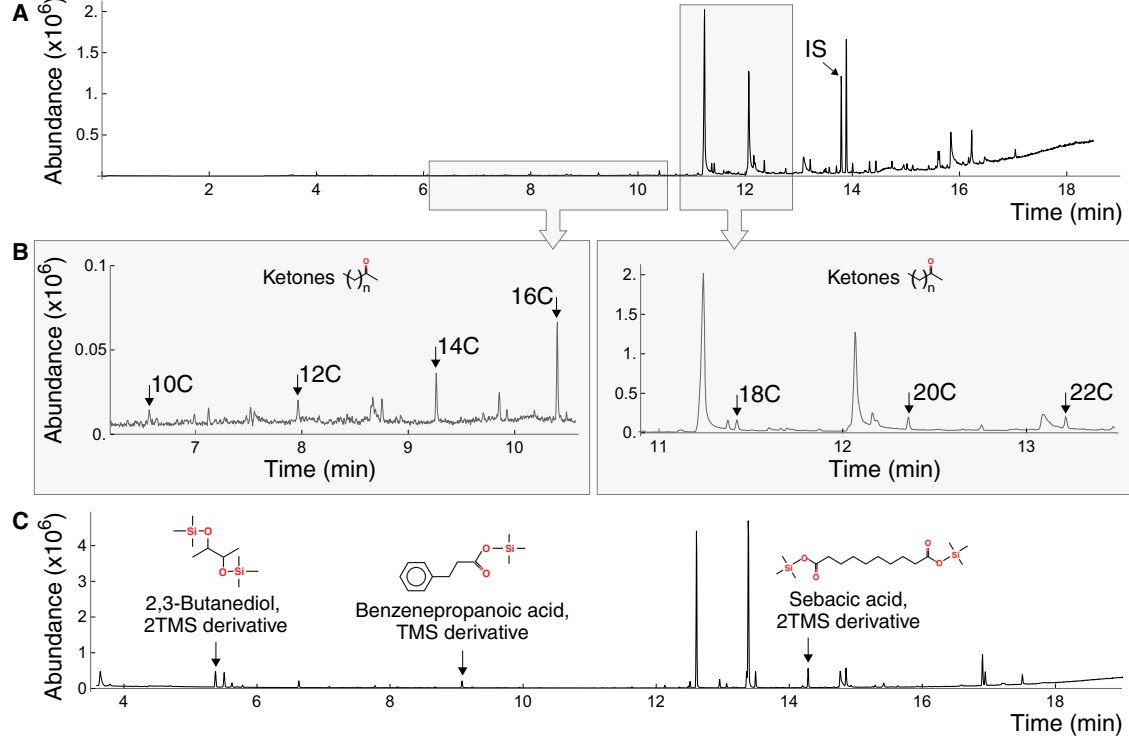

**Fig. 3 | Identification of PE degradation by-products via GC-MS.**
**A–C** Chromatograms of PE treated with GmSal, indicating different compounds. **A**, **B** Ketones of different length, indicated by the number of carbon atoms. **C** 2,3-Butanediol 2TMS derivative, benzenepropanoic acid TMS derivative, sebacic acid sTMS derivative. Compounds in C were identified with silylation (see "Methods"). IS: internal standard. Source data are provided as a Source Data file.

To further identify if the wide peak contained subspecies, an ion exchange chromatography (IEX) was run with a second saliva sample, which showed four well-defined elution peaks (peaks 1 to 4 in Supplementary Fig. 3B). Analysis by SDS-PAGE indicated that they all contained proteins of similar molecular weight (Supplementary Fig. 3D). To check which protein fractions of both the SEC and IEX retained PE degradation activity, aliquots of the eluted fractions were tested on a PE film. Using RAMAN spectroscopy, degradation activity was analyzed from fractions of the IEX four peaks (Fig. 4C–G) and in the SEC major peak (Supplementary Fig 4). Peaks indicated as 1, 2 and 3 (as in Supplementary Fig. 3B) showed PE oxidation. High degradation activity was also detected in the SEC main peak (peak 5 in Supplementary Fig. 3A) (Fig. S4).

Proteomics of IEX peaks 1, 2, and 3 revealed the presence of a handful of proteins, belonging to the arthropodan hexamerin/prophenoloxidase superfamily (peaks 1, 2, and 3, respectively)

(Supplementary Data 2). This result on the one hand confirmed the outcome of the SEC fraction proteomics (Supplementary Data 2, peak 5), and on the other refined it, reducing the number of potential candidates present in each peak. The fact that this family comprehends oxidase activities, made them the obvious candidates for PE degradation capacity within GmSal. These proteins, namely arylphorin subunit alpha, arylphorin subunit alpha-like, and the hexamerin acidic juvenile hormone-suppressible protein 1 were produced using a recombinant expression system and tested for this ability.

**Identification of wax worm enzymes as PE oxidizers**

In order to assess the activity of these proteins, 5 µl of each purified enzyme at a concentration of 1–5 µg/µl, were applied separately 8 sequential times (90 min each) on PE films. While arylphorin subunit alpha did not show any effect on PE film, arylphorin subunit alpha-like, re-named Demetra (NCBI accession number: XP_026756396.1), caused

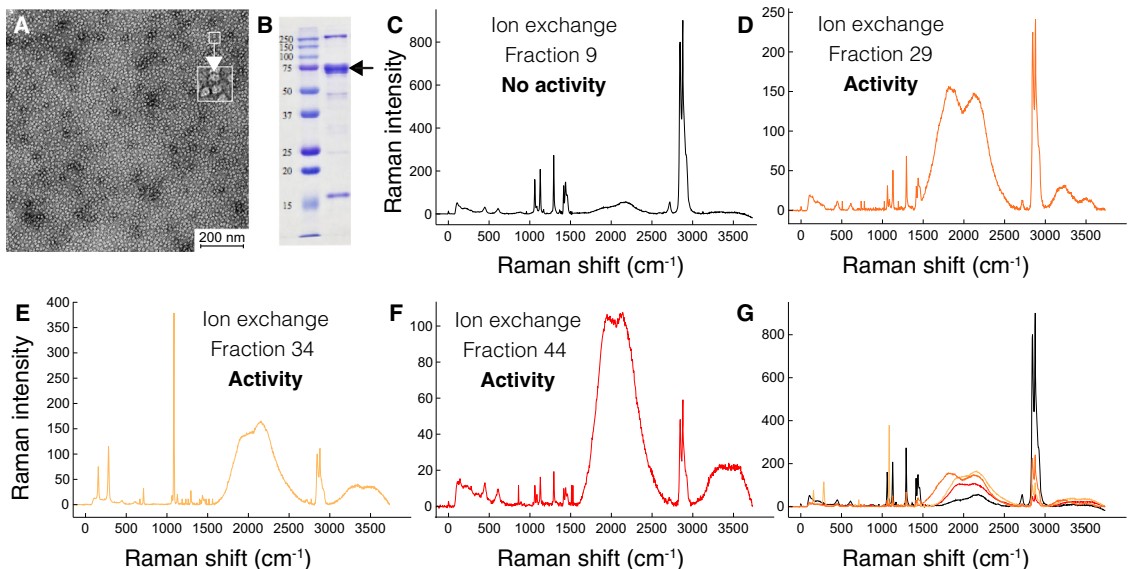

**Fig. 4 | GmSal content functional characterization. A** Electron microscopy negative staining of a saliva sample, dilution 1:500. Protein complexes (10–15 nm size) are indicated in the top right square. **B** SDS-PAGE of a GmSal sample, dilution 1:50. Molecular weight standards are listed on the left. The arrow indicates the major band at 75 kDa. **C–F.** RAMAN of PE treated with IEX fractions (see Fig. S4). **C** Fraction 9, no activity (control PE film, see Fig. 1 for details). **D–F** Profile corresponding to IEX peak 1 (fraction 29), 2 (fraction 34) and 3 (fraction 44) (see Fig. 1 for details). The intense peak at 1085 cm⁻¹ in E indicates amorphous PE. The decrease of the peaks at 2845 and 2878 cm⁻¹ indicates a less significant presence of PE. **G** Overlapping of fractions **C** to **F**. Experiments in figures **A**, **B** were performed multiple times (>5). Source data are provided as a Source Data file.

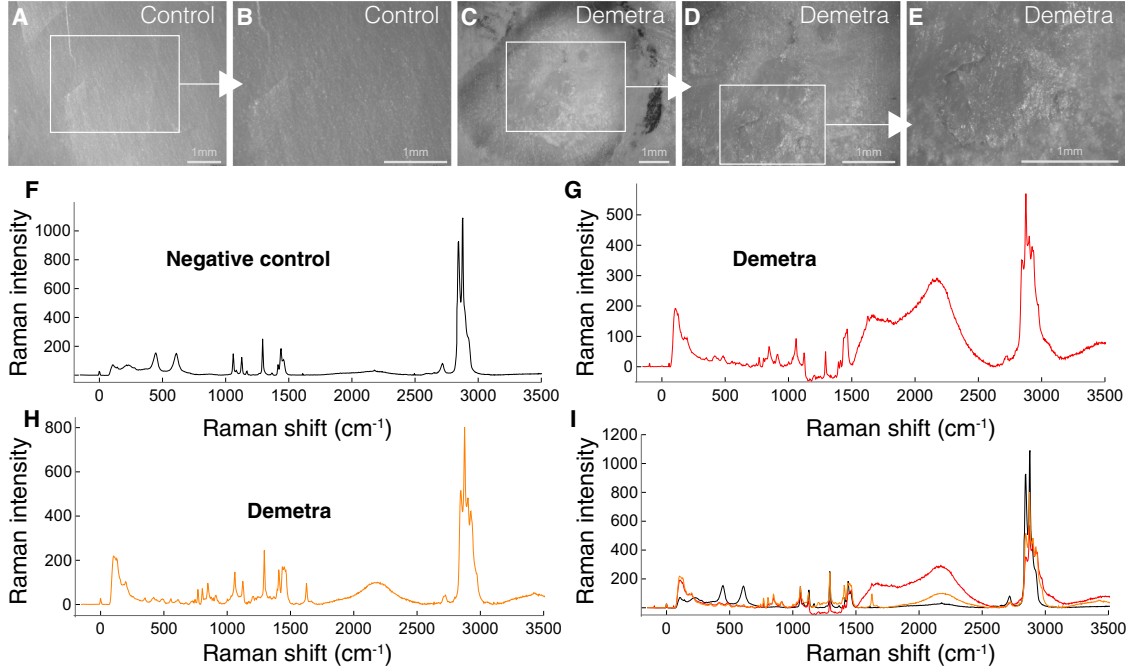

**Fig. 5 | Demetra effect on PE film. A**, **B** Control PE film. C-E. Demetra on PE film. **F–I** RAMAN spectroscopy of control (**F**) and Demetra treated PE film (**G**, **H**). **F** Control PE film (see Fig. 1 for details). **G**, **H** Two different punctual analyses within the crater as showed in **E**, indicating PE deterioration (same experimental conditions). See Fig. 1 for details. Moreover, the bands at 845 and 916 cm⁻¹ correspond to C–O–C and C–COO groups, respectively. The presence of PE is insignificant in **G**. Experiments in figures **A–E** were performed multiple times (>5). Source data are provided as a Source Data file.

PE deterioration with occasional conspicuous visual effect on the film itself (Fig. 5A–E and Supplementary Fig. 5). RAMAN confirmed oxidation and a damaged PE signature (Fig. 5F–H). The hexamerin, re-named Ceres (NCBI accession number: XP_026756459.1), caused PE oxidation/deterioration (Supplementary Fig. 6), but without the visual effect caused by Demetra. The same experiment performed with inactivated enzymes did not show any modification on the PE film, as indicated by

RAMAN analysis (Supplementary Fig. 7). The FTIR of PE treated with Demetra and Ceres also showed differences in the extent of oxidation between the two PEases (Supplementary Fig. 8). These results support the original hypothesis based on the idea that the buccal secretion is the main source of plastic degradation activities in the ww.

To analyze the potential of the saliva proteins in oxidizing PE, GC-MS was performed on PE granules (PE 4000) exposed to Demetra or

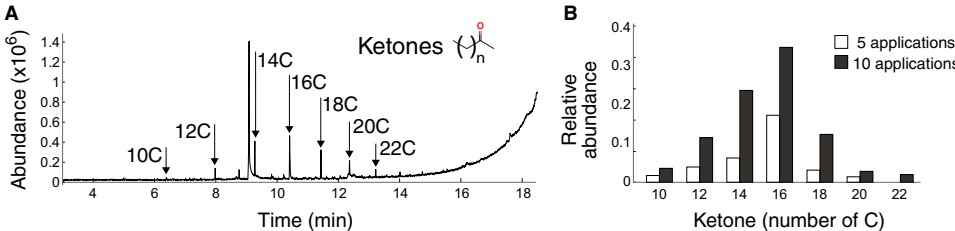

**Fig. 6 | Generation of PE degradation by-products by Demetra. A** GC-MS chromatogram of PE treated with Demetra. The arrows indicated the peaks corresponding to ketones with different number of carbons. **B** Increase of ketones formation as degradation products from five to ten applications of Demetra to PE. Source data are provided as a Source Data file.

Ceres. After 24 applications of Demetra (10 μl at 1.2 mg/mL, 90 min each), 2-ketones from 10 to 22 carbons were detected in the supernatant using GC-MS, with the fragment m/z of 58, and retention time used for identification (Fig. 6A). Increasing the treatment (ten versus five applications, 90 min each) showed an increase of 2-ketones of 10 to 20 carbons in relative abundance, and the appearance of 2-docosanone which was not detected after five applications (Fig. 6B). On the other hand, after PE 4000 treatment with Ceres, no by products were detected in the supernatant, suggesting substantial difference between the two proteins, despite the fact that they share the capacity to oxidize PE. Both enzymes present the same functional domains as haemocyanins' (Fig. S9A, B). However, the sequence comparison between the two enzymes shows only 30% of identity (Fig. S9C). Moreover, in silico analysis (Fig. S9D) indicates that Demetra is more stable than Ceres, which could be one of the reasons contributing to the observed differences.

## Discussion

This study reports that the saliva of the ww oxidizes and depolymerizes PE, with ww enzymes therein capable of reproducing the effect observed with the whole saliva. To the best of our knowledge, this is the first report of an enzymatic activity capable of attacking the PE polymer without any previous abiotic treatment. This capacity is achieved by animal enzymes working at room temperature and in aqueous solution with a neutral pH. Under these conditions, the enzymatic action of the ww saliva overcomes in a few hours a recognized bottleneck step (i.e., oxidation) in PE degradation[6,8–10].

The capacity of *Galleria mellonella* as well as other Coleoptera and Lepidoptera to degrade sturdy polyolefin-derived polymers as PE or PS has been extensively documented in the past few years, with no pretreatment required for the plastic polymer to be degraded[26–29,31–38,42–47]. If this capacity resides in the microorganisms of the worm gut, in the invertebrate itself, or in a complementation of the two, is still an object of debate. The gut microbiome has traditionally been considered the culprit of plastic degradation by insects. However, despite the numerous reports appeared lately about microbe species being potentially responsible for insect-driven plastic biodegradation, no consensus on specific species or genera of bacteria/fungi colonizing the Lepidoptera and Coleoptera gut and involved in plastic degradation has been reached[26–28,30,31,34–38,43]. Indeed, the exclusive involvement of microorganisms in this process has been recently questioned[32,48]. As for *G. mellonella*, the fast degradation makes it improbable for the gut microbiota to be the sole player in the observed modification of PE chemical structure, as already suggested[32]. This study provides solid evidence in favour of this argument and confirms the key role played by the larvae of *G. mellonella* in PE degradation: the saliva of the ww, and the enzymes it contains, oxidizes and depolymerizes PE. The action of the enzymes present in the saliva of *G. mellonella* on PE is therefore equivalent to that of abiotic pretreatments. The saliva-dependent oxidation of PE could thus provide a suitable substrate for further biological attack by causing the scission of the long polymer chains into smaller molecules that could then be metabolized and assimilated along the insect's digestive system (by the microbiome and/or by the insect's cells).

The current paradigm/hypothesis of PE biodegradation stands on the breaking of the C–C bond via the same mechanism that bacteria deploy to break alkanes[49,50]. The wax worms live and grow in the honeycombs of the beehives. They feed among other things (pollen, larvae, etc.) on beeswax. Given the similarity between plastics and beeswax, it is conceivable that the observed effect on PE is a consequence of the worm's capability to degrade wax. However, the two PEases, Demetra and Ceres, are an arylphorin and an hexamerin, respectively[51], and they are phylogenetically related to phenoloxidases (enzymes targeting aromatic rings) and hemocyanins, oxygen transport proteins that also present phenoloxidase activity[52,53]. In the literature diverse functions are attributed to this family of proteins, such as storage[54], immunity[55], and defence against plant phenolics[56]. The presence of this type of enzymes in the saliva of insects, already described in the literature[57–59], has a clear functional interpretation as the first line of defence against pathogens and plant anti-herbivore mechanisms[57–59]. The ability of insect enzymes to detoxify plant phenolics[56] suggests the potential existence of an alternative way to degrade PE. Polymeric chains are not the only components of plastics, as a number of small molecules, generically known as additives, are normally used to confer the desired properties to plastic objects. The appearance of a small aromatic compounds recognizable as a plastic additive such as benzenepropanoic acid, TMS derivative among the identified degradation products raises the possibility that this compound could become the target of the phenoloxidase-like activity of the ww enzymes. A consequence of this potential action of the ww enzymes on aromatic additives could be the formation of free radicals leading to the initiation of the autoxidative chain reaction[6]. The idea of the formation of free radicals as a first step to trigger autooxidation is not new, having been proposed as a way in which some bacteria might initiate plastic biodeterioration, via still unknown enzymes[14,20].

The ecological niche of *G. mellonella* larvae implies that they must deal with the presence in the beehive (wax, propolis, pollen...) of an extended variety of phenols[60–62]. This might provide an explanation to their capacity of PE degradation.

However, this will remain a speculation until the presence/formation of radicals is verified. Further studies will be required to get deeper insights into the functional modality of these ww enzymes, their diversity (as already indicated by the differences in the experimental outcomes between Demetra and Ceres), and the molecular mechanisms acting on PE.

Limited information is available about closely related proteins. A sequence comparison shows that both enzymes exhibit some degree of sequence identity with only a few proteins in the NCBI database (Supplementary Data 3 and 4). Of these proteins, only a handful (two arylphorins and one storage protein) have been structurally characterized to-date, and none of them shows a high degree of sequence identity with Demetra and Ceres (between 30 and 57%) (Supplementary Table 2). Further structural, biochemical, and functional studies will be necessary to have an exhaustive understanding of

the mechanisms of action of these PEases, the first of their kind to be described, to the best of our knowledge.

The existence of such enzymes, secreted from the mouth and evolved to work at room temperature and neutral pH on plastic, provides a promising alternative paradigm for the biological degradation of PE. This framework goes beyond the current definition of biodegradation, which is exclusively based on the conversion of plastic to $CO_2$ through the metabolic activity of microorganisms. On one hand, the observed oxidation and deterioration of PE does not depend on any microbial activity; on the other hand, the easy working conditions and the appearance of degradation products such as ketones and additives suggest the potential use of these enzymes for plastic waste degradation and recycling or upcycling of plastic components. This could be used either as an alternative to the metabolic conversion of plastic to $CO_2$, or as the initial oxidative step in combination with standard microbial degradation pathways.

Further, this study suggests that insect saliva might result as a depository of degrading enzymes (plastic, cellulose or lignin to mention some) which could revolutionize the bioremediation field. Although further studies will be necessary to obtain a deeper understanding of the step-by-step evolution of plastic in contact with ww saliva enzymes, this discovery introduces another potential approach for dealing with plastic degradation. In a circular economy frame, this study opens up a potential field both in plastic upcycling, and in manufacturing the plastic of the future, with ad-hoc formulations prone to facilitate degradation by selected enzymes.

## Methods

### Resource availability
Further information and requests for resources and reagents should be directed to and will be fulfilled by the lead contact, Federica Bertocchini (federica.bertocchini@csic.es).

### Experimental model and subject details
*Galleria mellonella* larvae colony was maintained in an incubator at 28 °C in the dark, and fed with beeswax from beehives.

### Wax worm saliva collection
Larvae of 150–300 mg were used for saliva collection. Briefly, a glass capillary connected to a mouth pipet was placed at the buccal opening and the liquid was collected. Five to ten microliters of saliva were collected from each worm. The concentration of proteins in the saliva measured via Bradford methodology varies between 20 and 30 mg/ml. Saliva for PE application was immediately used. Occasionally, frozen saliva-only can be utilized. For electron microscopy, saliva was diluted 1:1 in the following buffer: 10 mM Tris-Cl pH 8, 50 mM NaCl. For proteomic analyses, saliva was diluted 1:1 in 10 mM Tris-Cl pH 8, 50 mM NaCl, 2 mM DTT, 20% glycerol. For control, *Samia cynthia* larvae (kindly provided by InsectPark-Microfauna S.L. (El Escorial, Madrid)) at the last stage were used to collect saliva as described above.

### RAMAN and FTIR analyses
PE film was treated with 30 µl of GmSal for 90 min, three applications for RAMAN. PE film was treated with 5 µl of GmSal for 90 min, nine applications for FTIR. Recombinant proteins were applied as follows: 5 µl of protein (concentration between 1 and 5 µg/ml) were applied eight times on PE film 90 min each time, 16 applications for FTIR. For the control with inactivated proteins, recombinant proteins were denatured at 100 degrees centigrade for 10 min. SEC and IEX peak aliquots were applied six times, 30 min each, and left overnight. Treated and control films were washed with water and ethanol. RAMAN analyses were performed on (treated and control) PE films using Alpha300R–Alpha300A AFM Witec equipment with 5 mW power, 50× (NA0.8) objective, integration time 1, accumulation 30, wavelength 532 nm. FTIR analyses were performed with a Jasco LE-

4200 equipment, with the following features: interval 4000–400 cm⁻¹, Resolution 4 cm⁻¹, scan 264.

### High temperature-gel permeation chromatography (HT-GPC) analysis
HT-GPS was performed by Polymer Chart, Valencia, Spain.

Briefly, the experimental conditions are the following: Equipment: GPC-IR5_I Polymer Char; solvent: TCB stabilized with 300 ppm of BHT; dissolution temperature, detectors temperature, columns temperature: 160°; volume: 8 ml; weight: 8 mg; dissolution time: 60 min; injected volume: 200 µl; injection time: 55 min; flow: 1 ml/min; columns: 3 × PL gel Olexis Mix-Bed columns (13 microns), 300 × 7.5 mm + guard column

For the carbonyl index analysis, GPC_IR6 was used, with dissolvent o-DCB and temperature at 150°.

PE film and PE 4000 (20 mg) were treated with 100 µl of GmSal for 90 min. The treatment was repeated 15 times (film and PE 4000) and 30 times (PE 4000).

### Gas Chromatography Mass Spectroscopy (GC-MS) and Tandem analysis
An amount of 20 mg of PE 4000 was placed in a 1.5 ml Eppendorf tube. PE was exposed to 40 µl of *G. mellonella* saliva 9 times for 90 min each at room temperature and avoiding light. For prolonged treatments (days 1, 2, 3, and 6), three applications of 100 µl of saliva for 90 min each at room temperature were carried out each day. PE controls were performed using Milli-Q water in substitution of the saliva of *G. mellonella* larvae, as well as saliva of *G. mellonella* larvae only. Also, PE was exposed to 10 µl (1.2 mg/mL) of Demetra 24 times for 90 min. Prolonged treatment was performed as well for Demetra (days 1 and 2), five applications per day of 10 µl (1.2 mg/mL) for 90 min each. As control, the same experiment was repeated using the protein buffer. Afterward, samples were centrifuged with an Eppendorf centrifuge 5810R at 19083 × *g* for 30 s and the subnatant was transferred to a new 1.5 ml Eppendorf tube. Samples and controls were extracted using a QuEChERS (quick, easy, cheap, effective, and safe) method[63] based on[64] with some modifications. Briefly, 50 µl of diphenyl phthalate (Internal Standard; IS) at a concentration of 1 mg/ml was added at each sample and extracted with 300 µl of dichloromethane (DCM) and 5% (v/m) of NaCl. The tube was vortexed for 30 s and sonicated in a bath (50/60 Hz) for 15 min at room temperature, followed by centrifugation with an Eppendorf centrifuge 5810R at 20 °C and 19083 × *g* for 10 min. Finally, DCM located as the subnatant was collected and placed in an insert before analysis. Silylation reaction with N, O-Bis(trimethylsilyl) trifluoroacetamide (BSTFA) was performed to determine the low-volatility polar compounds which show low detection sensibility. A fraction of 50 µl of each sample with 50 µl of BSTFA was incubated for 20 min at 60 °C before the analysis.

Dichloromethane (DCM; CAS-No: 75-09-2) for gas chromatography-mass spectrometry (GC-MS) was SupraSolv grade purity and obtained from Sigma-Aldrich (Darmstadt, Germany). Sodium chloride (NaCl; ≥ 99.5%; CAS-No: 7647-14-5) and ultrapure water from a Milli-Q system were supplied from Merck (Darmstadt, Germany).

Crystalline granular powder polyethylene (PE 4000; CAS-No: 9002-88-4, specification sheet in https://www.sigmaaldrich.com/ES/es/product/aldrich/427772) was supplied by Sigma-Aldrich (Saint Louis, USA).

Chromatographic analyses were performed with a gas chromatography-mass spectrometry system (GC-MS) 7980A-5975C from Agilent Technologies. Separation of the metabolites was performed on a DB-5th Column coated with polyimide (30 m length, 0.25 mm inner diameter, and 0.1 µm film thickness; Agilent Technologies, USA) for proper separation of substances, and Helium (He) was utilized as a carrier gas. The analysis was performed using a split

injector at 350 °C and an injection volume of 1 μl. The ion source temperature was 230 °C, the °C mass spectral analysis was performed in scan mode, the quadrupole temperature of 150 °C, and a fragmentation voltage of 70 eV. The oven programme started at 60 °C for 3 min, then 20 °C/min to 350 °C for 1 min. The total run time was 18.5 min and 19.5 min for derivatized samples. The resulting chromatograms were processed using the software MSD ChemStation E.01.00.237 from Agilent Technologies, Inc while for the identification NIST11 library was used.

The evaluation of the prolonged treatment was based on the relative abundance of each untargeted compound, which consists of the quotient of the area under the peak of each compound divided by the area under the peak of the IS.

Gas Chromatography/Tandem Mass Spectrometry was used for confirmation of the non-target compounds by a BRUKER 456-GC SCION TQ. This experiment was performed by the Elemental and Molecular Analysis facility, University of Extremadura, Spain. Briefly, the injector port was set at 230 °C in Split mode. Separation was achieved using a column HP 5MS, 30 m, 0.25 mm, and 0.25 μm. Helium (He) was utilized as a carrier gas. The column oven was programmed in the following conditions: 60 °C for 3 min, increase of 20 °C/min to 325 °C for 1 min. The collision energy was 15 eV.

### Electron Microscopy analysis

Larvae saliva samples were diluted 1:50 in the proper buffer (see "Wax worm saliva collection").

Samples were analyzed by electron microscopy (EM) after being adsorbed to glow-discharged carbon coated grids and stained with 2% uranyl acetate. Grids were observed using a JEOL JEM-1230 EM operated at 100 kV and a nominal magnification of 40000. EM images were taken under low dose conditions with a CMOS Tvips TemCam-F416 camera, at 2.84 Å per pixel.

### Protein Chromatography analyses

For the size exclusion chromatography, 40 μl of ww saliva in the proper buffer (see "Wax worm saliva collection") was thawed, pooled, and centrifuged. The supernatant was filtered (0.45 μm cutoff, Ultrafree Millipore) and loaded to a size exclusion chromatography column Superdex 200 5-150 (Cytiva) equilibrated with 10 mM Tris-Cl, 50 mM NaCl, 2 mM DTT.

For the ion exchange chromatography, upon thawing, the sample was diluted to 100 μl with 10 mM Tris-Cl at pH 8, centrifuged, filtered and the supernatant loaded to a monoQ 5/50 GL ion exchange column (Cytiva). After a wash step, a 40 ml gradient with buffer A (10 mM Tris-Cl pH8), and buffer B (same as A supplemented with 500 mM NaCl) was applied.

### Proteomic analysis

Liquid Chromatography Mass Spectrometry (LC-MS) analysis. All peptide separations were carried out on an Easy-nLC 1000 nano system (Thermo Fisher Scientific). For each analysis, the sample was loaded into a precolumn Acclaim PepMap 100 (Thermo Fisher Scientific) and eluted in a RSLC PepMap C18, 15 cm long, 50 μm inner diameter and 2 μm particle size (Thermo Fisher Scientific). The mobile phase flow rate was 300 nl/min using 0.1% formic acid in water (solvent A) and 0.1% formic acid and 100% acetonitrile (solvent B). The gradient profile was set as follows: 5–35% solvent B for 45 min, 35–100% solvent B for 5 min, 100% solvent B for 10 min. Four μl of each sample were injected.

MS analysis was performed using a Q Exactive mass spectrometer (Thermo Fisher Scientific). For ionization, 1900 V of liquid junction voltage and 270 °C capillary temperature was used. The full scan method employed a m/z 400–1500 mass selection, an Orbitrap resolution of 70,000 (at m/z 200), a target automatic gain control (AGC) value of $3^{e6}$, and maximum injection times of 100 ms. After the survey scan, the 15 most intense precursor ions were selected for MS/MS fragmentation. Fragmentation was performed with a normalized collision energy of 27 eV and MS/MS scans were acquired with a starting mass of m/z 100, AGC target was $2^{e5}$, resolution of 17,500 (at m/z 200), intensity threshold of $8^{e4}$, isolation window of 2 m/z units and maximum IT was 100 ms. Charge state screening was enabled to reject unassigned, singly charged, and equal or more than seven protonated ions. A dynamic exclusion time of 20 s was used to discriminate against previously selected ions.

MS data analysis. Mass spectra *.raw files were searched against an in-house specific database against *Galleria_Proteins* (12715 proteins entries) extracted from the genome annotation available in Mendeley (https://doi.org/10.17632/t7b5s58vxt.3), using the Sequest search engine through Proteome Discoverer (version 1.4.1.14) (Thermo Scientific). Search parameters included a maximum of two missed cleavages allowed, carbamidomethyl of cysteines as a fixed modification and oxidation of methionine as variable modifications. Precursor and fragment mass tolerance were set to 10 ppm and 0.02 Da, respectively. Identified peptides were validated using the Percolator algorithm with a *q*-value threshold of 0.01. The protein identification by nLC-MS/MS was carried out in the Proteomics and Genomics Facility (CIB-CSIC), a member of ProteoRed-ISCIII network[65]. The mass spectrometry proteomics data have been deposited to the ProteomeXchange Consortium via the PRIDE[66] partner repository with the identifiers PXD035476 and PXD035479.

### Recombinant protein production and utilization

Arylphorin, arylphorin subunit alpha-like (Demetra) and hexamerin (Ceres) were produced by Genscript, utilizing the baculovirus expression system in insect cells, according to the manufacturer. Briefly, Sf9 cells were infected with P2 baculovirus, flasks were incubated at 27 °C for 48–72 h and media harvested. Then cells were removed, and transfection medium was applied for purification. The produced proteins were resuspended in 150 mM NaCl, 20 mM HEPES, 5% glycerol and used for the degradation assay. The same buffer alone was used as negative control.

### Galleria mellonella genome annotation

In order to obtain the most useful information from mass spectrometry analysis of proteins extracted from ww saliva, it is pivotal to use a representative database of protein sequences. In addition to the NCBI official annotation of *Galleria mellonella* protein sequences, a new annotation was produced exploiting also the information of *Galleria mellonella* salivary glands RNA-seq data. *Galleria mellonella* genome annotation was performed using the genome sequence available at NCBI. A specific pipeline was developed to combine the information from RNA-seq with ab initio predictors in order to obtain the most accurate annotation. Briefly, the RNA-seq data was mapped on the reference genome using STAR (version 2.5 0c)[67] in local mode and used to perform a reference-guided transcriptome assembly with Trinity (v2.11.0)[68]. The obtained transcripts and the mapping files were used as input for the Braker2 pipeline[69] to combine AUGUSTUS ab initio annotation[70] with the transcriptome assembly to obtain the annotation in GFF format, together with transcript and protein sequences. Proteins were used as input for the PANNZER2 pipeline to obtain descriptions[71], Gene Ontology and KEGG annotations. About 32000 genes could be annotated in the Galleria genome. The corresponding proteins were analyzed to assess their completeness performing a BLASTP alignment against the UniRef90 database and calculating the percentage of alignment. A similarity search against the UniRef90 (Nov2018) database showed that about 50% of the predicted proteins covered 100% of the corresponding hits (i.e., full length) and that about 80% of the predicted proteins covered at least 50% of the corresponding hits. As a further control, the proteins and the genome were evaluated with the BUSCOv2 pipeline[72]. The BUSCO database

contains sets of single-copy highly conserved genes across different taxa (i.e., Eukaryota or Insects). By performing an analysis with the BUSCO database it is possible to assess the completeness of a genome/proteome, the presence of duplications and/or fragmentations. This analysis was performed using the predicted proteome and also the unannotated genome for a comparison. By comparing the results of the unannotated with the annotated genome, we can see a small fraction of missing genes which are probably absent from the genome assembly and that cannot be recovered from the current genome sequence. This explains some missing genes present in the later NCBI annotation.

The RNA-seq data that was used for the G. mellonella genome annotation has been deposited in NCBI under the BioProject accession number PRJNA822887. In addition, the newly generated annotation in GFF3/GTF format was deposited to Data Mendeley (https://doi.org/10.17632/t7b5s58vxt.3)

### Analyses of the wax worm PEases sequences

The domain composition of the proteins was predicted with the SMART algorithm[73] selecting the options: Outlier Homologues, PFAM, signal peptides, and internal repeats. In order to retrieve similar proteins with a resolved 3D structure, a BLAST search was performed on the PDB database[74] using an e-value cutoff of 0.001 for both the proteins, whereas a minimum sequence identity of 50% was selected for Demetra (XP_026756396.1) and 30% for Ceres (XP_026756459.1). The two protein sequences were aligned using Clustal Omega[75] and then visualized with Jalview[76]. Finally, the prediction of the biochemical characteristics of the two proteins was performed with Expasy's ProtParam tool[77].

### Statistics and reproducibility

No statistical methods were used in this work. No statistical method was used to predetermine sample size. No data were excluded from the analyses.

### Reporting summary

Further information on research design is available in the Nature Research Reporting Summary linked to this article.

## Data availability

The data used in the figures are available in the Source Data and Source Data Supplementary Material files. The RNA-seq data that was used for the G. mellonella genome annotation have been deposited in NCBI under the BioProject accession number PRJNA822887. In addition, the newly generated annotation in GFF3/GTF format was deposited in Data Mendeley (https://doi.org/10.17632/t7b5s58vxt.3). The mass spectrometry proteomics data have been deposited to the ProteomeXchange Consortium via the PRIDE partner repository with the dataset identifiers PXD035476 and PXD035479. Source data are provided with this paper.

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

## Acknowledgements

We thank the Proteomics and Genomics, Electron Microscopy and Gas Chromatography facilities of the CIB for the excellent technical support. We thank Paloma Delgado and Pedro Velasco from InsectPark-Microfauna S.L., El Escorial, Madrid, Spain (https://insectpark.es/) for providing *Samia cynthia* larvae. We thank the Centres Científics i Tecnològics Unitat Cromatografia de Gases-Espectrometria de Masses Aplicada (CCIT), University of Barcelona, Spain for the support provided in the GC-MS experiments. We thank the Roechling foundation for supporting and sponsoring this work. This work has been funded as follows: Roechling Stiftung to F.B., Consejo Superior de Investigacion Cientifica (CSIC) to F.B., NATO Science for Peace and Security Programme (Grant SPS G5536) to T.T., Junta de Castilla y León, Consejería de Educación y Cultura y Fondo Social Europeo (Grant BU263P18) to T.T., Ministerio de Ciencia e Innovación (Grant PID2019-111215RB-100) to T.T., The Generalitat de Catalunya (2017 SGR 1192) to M.S., and Ministerio de Ciencia e Innovación (Grant PID2020-120275GB-I00) to E.A.-P.

## Author contributions

Conceptualization: F.B. and C.F.A. Methodology: A.S.-V., P.C.-V., F.R.-V., M.B.-V., M.S.-A., E.R.-L., R.I.-V., P.C., R.A.C., M.M., P.F., C.P., L.G.-L., E.A.-P., M.S., T.T., C.F.A., and F.B. Investigations: A.S.-V., P.C.-V., F.R.-V., M.B.-V., M.S.-A., E.R.-L., R.I.-V., P.C., R.A.C., M.M., P.F., C.P., L.G.-.L, E.A.-P., M.S., T.T., C.F.A., and F.B. Visualization: F.B., C.F.A., P.C.V., L.G., M.S., and E.A.-P. Funding acquisition: F.B., T.T., M.S., and E.A.-P. Project administration: F.B. Writing—original draft: F.B. Writing—review & editing: A.S.-V., P.C.-V., F.R.-V., M.B.-V., M.S.-A., E.R.-L., R.I.-V., P.C., R.A.C., M.M., P.F., C.P., L.G.-L., E.A.-P., M.S., T.T., C.F.A., and F.B.

## Competing interests

The authors declare no competing interests.
