## [Peer Review File · Nature Communications]

REVIEWER COMMENTS

Reviewer #1 (Remarks to the Author):

This is a very interesting paper and its publication will make a significant contribution to the literature. However, there were several issues that need to be addressed before the manuscript may be considered ready for publication.

General Comments: The paper discusses the problem with mechanical and chemical treatments of polyethylene (PE) waste, to introduce microbial degradation of PE. Microbial degradation is still in its early stages, therefore, there is a lot of unknown aspects that need to be addressed. One of the key steps in the degradation process is the oxidation of PE prior to the bio-assimilation or mineralization steps. By observing the changes in PE film when it was exposed to wax worm saliva secretion, it is suggested that there were unknown enzyme activities, which were capable of biodegrading PE. The paper confirmed changes in PE integrity before and after saliva treatment, confirming the idea that wax worm saliva oxidizes PE film to form carbonyl groups. The formation of smaller molecules after saliva treatment as well as a slight changes in the average molecular weight of the treated PE indicated that wax worm saliva has an oxidative effect on PE film and can depolymerize PE into smaller fragments. The smaller fragments detected were identified as oxidized aliphatic chains such as 2-ketones, comprised of 10 to 22 carbon atoms. The identities of other by-products were also found. Next, the proteins present in the wax worm saliva were found to belong to the hexamerin/prophenoloxidase superfamily. There are two important proteins, named Demetra and Ceres, which were identified as an arylphorin subunit alpha-like protein and a hexamerin protein, correspondingly. The enzymatic activities of these two enzymes were tested on PE film to confirm that they can break down PE. Both enzymes were able to cause biodeterioration on PE film, with Demetra capable of inducing visual evidence of break-down. Further investigations on the enzyme activities showed that only Demetra produced 2-ketones (from 10 to 22 carbons) upon exposure to PE granules. Thus, Demetra and Ceres both have oxidative effect on PE, but the extend of the oxidation varied.

Validity: The paper follows a step-by-step strategy of experiments, with each experimental results supporting the previous one. Additionally, the paper data interpretation was very clear for each of the results.

Significance: The result of this paper is novel in term of identification of unknown secreted enzymes that are capable of oxidizing PE like that using abiotic factors but with a shorter time frame. The identification of these enzymes will help speed-up the biodegradation rate tremendously and remove the bottleneck of the PE degradation process. However, Demetra and Ceres had similar oxidation activities on PE, but the effects were different, shown in the physical deterioration of PE as well as the by-products identified. A more comprehensive comparison between the two should be done including

chemical structure of the two, substrate specificity, sequence comparison and other biochemical characterization. The results will help explain the present and function of Ceres, since Demetra alone does show the same if not better results than Ceres. A more in-depth knowledge about Demetra and Ceres can aid in other types of bioplastic degradation process as the enzymes may have similar oxidative affect. Moreover, the identification of Demetra and Ceres in wax worm saliva opens a new possible area of study which involve secreted oxidation enzymes. The present of secreted enzymes in the wax worm saliva raises the question of origin of the enzymes, whether it is secreted from the larvae themselves because of evolutionary adaptations or due to the present of the microbiome in their gut.

Clarity and context: The experiments were conducted appropriately for each point of interest. The experiment steps and data were easy to follow and data interpretation for each step were clear and sufficient. The overlapping data representation show clearly the changes before and after saliva treatment.

References: The paper cites previous literature appropriately.

Providing constructive feedback: Overall, the paper was easy to follow, with straight forward experiment goals and data interpretation. It may be helpful to include a clear statement of the hypotheses that underlay the research objectives in the introduction part. However, the introduction clearly stated the problem with plastic accumulation, different methods to deal with it, the process of wax worm become the center of the project, the summary of experimental process, and the identity of the two enzymes found.

Specific Comments: On page 6, the authors state: "To narrow down the number of potential candidates, a saliva sample was analyzed by size exclusion chromatography (SEC)."

- What was the volume of the ww saliva samples? What was the protein concentration of the ww saliva samples?

On page 6, the authors state: "More than 200 proteins were detected, including a variety of enzymatic activities, transport and structural proteins, etc (not shown)".

- How were the 200 proteins detected? and how were the functional activities identified? This should be explained briefly with details provided in the Methods section or in the Supplementary materials

On page 6, the authors state: “SDS-PAGE gel of the major fraction showed a strong band at about 75kDa (Supplementary Fig. S3C).”

- How was the 75 KDa band analyzed? Details of this should be included in the Methods section or in the Supplementary materials.

On page 7, in the legend of Figure 5:

- The legend of Figure 5 does not explain the difference in experimental conditions for Figure 5G versus Figure 5H. Similarly, in Supplementary Figure S6B, what is the different in terms of the experimental conditions between the upper and lower spectra?

On page 7, related to Figure 5:

- With respect to the amount of saliva used to treat PE film versus PE 4000 granules, how much PE were used per saliva per treatment was used for the RAMAN, FTIR and HT-GPC analyses? Does the amount of PE:Saliva ratio affect the efficiency of enzymes?

On page 9, in the Discussion: The authors state, “Unexpected instead is the capacity of these particular proteins to oxidize PE, a polymeric, compact hydrophobic substance. However, the ecological niche of Lepidoptera and the potential necessity to react to plant phenols might provide a possible explanation.”

- This part of the discussion needs to be revised. It seems that the authors do not know the biology of the Greater Waxmoth, *G. mellonella*. It is no surprise that larvae of *G. mellonella* secrete enzymes that can degrade hydrocarbons because *G. mellonella* larvae live in beehives and consume beeswax. This should be incorporated into the discussion.

On page 15, the authors state: “Larvae of 150-300 mg were used for saliva collection. Briefly, a glass capillary connected to a mouth pipet was placed at the buccal opening and the liquid was collected.” While the authors provide information about the mass of the larvae used, they do not provide any information about the volume of saliva collected at any one time, or the protein concentration in the saliva samples.

On page 17, the authors state: “Mass spectra *.raw files were searched against an in-house specific database against Galleria proteins (12715 proteins entries),...”.

- What is the nature of the “Galleria protein database”? Was this derived from an annotated genome sequence of *G. mellonella*? Is there a url link to this data base?

Suggested improvements: This reviewer suggests running FTIR on the PE treated with Demetra and Ceres separately to compare with the very first FTIR analysis of wax worm saliva on PE film. When Demetra and Ceres are present together, the FTIR spectra shows peaks that represent carbonyl-groups. Demetra by itself can show visual deterioration of PE, while Ceres does not. An additional FTIR analysis would show the generation of the carbonyl-groups from wax worm saliva is due to either Demetra or Ceres. In addition, kinetics of PE degradation by Demetra and/or Ceres with respect to protein quantity (mass) vs time would be very informative.

Reviewer's expertise: This reviewer has extensive expertise in the microbial and enzymatic degradation of PE and other natural and synthetic polyester polymers.

Reviewer #2 (Remarks to the Author):

The paper from Sanluis-Verdes, Colomer-Vidal, and colleagues investigates the enzymes found in wax worm saliva for polyethylene degradation. Overall, this is an incredibly interesting paper and I strongly endorse this study for urgent publication. This work will be extremely interesting to many in the research community focused on plastics biodegradation.

The authors, if room permits, should remove the sentence about “the first report of ...” and use that extra space to instead state what the enzymes are.

The sentence “[i]n fact, no such enzyme has been identified yet, confirming the crucial limiting role of oxidation in the whole biodegradation process chain” is an illogical statement to me. Because an enzyme has not been identified does not necessarily mean that oxidation is the crucial limiting role – this should be rewritten. I will note that I fully agree that oxidation is critical here – this is merely a comment on the logic flow of the sentence.

Why do the authors name the enzymes Demetra and Ceres? This is seemingly a reference to Greco-Roman mythological characters, but the enzyme names do not convey any useful information. The authors should use a more appropriate and informative name.

What does “physical deterioration of PE” mean?

Figure 1 – By “picks” in the figure caption, I assume the authors mean “peaks”? Maybe the authors could show the actual films themselves as they go through treatment? That would be potentially really nice to see.

I am wondering about the experimental protocols. Why was 90 minutes chosen as a reaction time and why was fresh enzyme or ww saliva then added? Did the authors run at much longer times with a single inoculum? In their experimental setup, did they dilute the ww saliva?

For the 6-day experiment, did the authors use HT-GPC to measure the MW distributions as well? That could be quite interesting. What is the extent of measurable product release in these experiments? That would be nice to quantify on a C-mole basis from the original substrate if possible.

I think that the authors would benefit the community if they were to include their full proteomics dataset in the manuscript, or (better yet) upload those data to a web server that is (post-publication) publicly accessible. I suspect that Nature Communications requires that anyway.

Are the enzymes in the arylphorin sub-domains of an intact enzyme, by chance?

How did the authors inactivate the enzymes in the referenced control experiment?

The lack of protein characterization here is quite deflating. This would make this paper (in my opinion) so much better, and seems like an obvious missed opportunity. At an absolute minimum, the authors should do a BLAST search to look for similar enzymes that have been characterized biochemically and/or structurally to see if these types of putative activities are unprecedented or not. The authors allude to the fact that others will need to follow-up on this discovery – certainly many will do so, but I think the authors could do further service by including at least literature review around these enzymes and closely-related homologues.

I think the authors should tone down or remove the “no pretreatment” statement – if animals are chewing the polymers, that’s pretreatment.

Did the authors attempt to express the enzymes in E. coli?

Reply to REVIEWER COMMENTS

We thank the Reviewers for their positive and encouraging comments.

We modified the manuscript according to the Reviewers' suggestions, as detailed below (in blue).

Reviewer #1 (Remarks to the Author):

We thank Reviewer 1 for his/her positive remarks. We modified the text according to the indicated suggestion, as detailed below.

This is a very interesting paper and its publication will make a significant contribution to the literature. However, there were several issues that need to be addressed before the manuscript may be considered ready for publication.

General Comments: The paper discusses the problem with mechanical and chemical treatments of polyethylene (PE) waste, to introduce microbial degradation of PE. Microbial degradation is still in its early stages, therefore, there is a lot of unknown aspects that need to be addressed. One of the key steps in the degradation process is the oxidation of PE prior to the bio-assimilation or mineralization steps. By observing the changes in PE film when it was exposed to wax worm saliva secretion, it is suggested that there were unknown enzyme activities, which were capable of biodegrading PE. The paper confirmed changes in PE integrity before and after saliva treatment, confirming the idea that wax worm saliva oxidizes PE film to form carbonyl groups. The formation of smaller molecules after saliva treatment as well as a slight changes in the average molecular weight of the treated PE indicated that wax worm saliva has an oxidative effect on PE film and can depolymerize PE into smaller fragments. The smaller fragments detected were identified as oxidized aliphatic chains such as 2-ketones, comprised of 10 to 22 carbon atoms. The identities of other by-products were also found. Next, the proteins present in the wax worm saliva were found to belong to the hexamerin/prophenoloxidase superfamily. There are two important proteins, named Demetra and Ceres, which were the identified as an arylphorin subunit alpha-like protein and a hexamerin protein, correspondingly. The enzymatic activities of these two enzymes were tested on PE film to confirm that they can break down PE. Both enzymes were able to cause biodeterioration on PE film, with Demetra capable of inducing visual evidence of break-down. Further investigations on the enzyme activities showed that only Demetra produced 2-ketones (from 10 to 22 carbons) upon exposure to PE granules. Thus, Demetra and Ceres both have oxidative effect on PE, but the extend of the oxidation varied.

Validity: The paper follows a step-by-step strategy of experiments, with each experimental results supporting the previous one. Additionally, the paper data interpretation was very clear for each of the results.

Significance: The result of this paper is novel in term of identification of unknown secreted enzymes that are capable of oxidizing PE like that using abiotic factors but with a shorter time frame. The identification of these enzymes will help speed-up the biodegradation rate tremendously and remove the bottleneck of the PE degradation process. However, Demetra and Ceres had similar oxidation activities on PE, but the effects were different, shown in the physical deterioration of PE as well as the by-products identified. A more comprehensive comparison between the two should be done including chemical structure of the two, substrate specificity, sequence comparison and other biochemical characterization. The results will help explain the present and function of Ceres, since Demetra alone does show the same if not better results than Ceres. A more in-depth knowledge about

Demetra and Ceres can aid in other types of bioplastic degradation process as the enzymes may have similar oxidative affect. Moreover, the identification of Demetra and Ceres in wax worm saliva opens a new possible area of study which involve secreted oxidation enzymes. The present of secreted enzymes in the wax worm saliva raises the question of origin of the enzymes, whether it is secreted from the larvae themselves because of evolutionary adaptations or due to the present of the microbiome in their gut.

Clarity and context: The experiments were conducted appropriately for each point of interest. The experiment steps and data were easy to follow and data interpretation for each step were clear and sufficient. The overlapping data representation show clearly the changes before and after saliva treatment.

References: The paper cites previous literature appropriately.

Providing constructive feedback: Overall, the paper was easy to follow, with straight forward experiment goals and data interpretation. It may be helpful to include a clear statement of the hypotheses that underlay the research objectives in the introduction part. However, the introduction clearly stated the problem with plastic accumulation, different methods to deal with it, the process of wax worm become the center of the project, the summary of experimental process, and the identity of the two enzymes found.

Specific Comments: On page 6, the authors state: “To narrow down the number of potential candidates, a saliva sample was analyzed by size exclusion chromatography (SEC).”

- What was the volume of the ww saliva samples? What was the protein concentration of the ww saliva samples?

The saliva volume was 40 microliters as specified in the Materials and Methods, section “Protein Chromatography analyses”, line 118.

The concentration of proteins in the saliva varies between 20 and 30 mg/ml. We added this information in the Materials and Methods, section “Wax worm saliva collection”, line 26.

On page 6, the authors state: “More than 200 proteins were detected, including a variety of enzymatic activities, transport and structural proteins, etc (not shown)”.

We are grateful for the Reviewer’s suggestion. We added the protein list in Supplementary table S2 in the supplementary material.

- How were the 200 proteins detected? and how were the functional activities identified? This should be explained briefly with details provided in the Methods section or in the Supplementary materials

The detection of the proteins in the saliva is explained in the section “Proteomic analysis” from Materials and Methods. In particular, in line 147, we described how the proteins have been identified (MS data analysis. Mass spectra *.raw files were searched against an in-house specific database against *Galleria mellonella*_Proteins (12715 proteins entries), using the Sequest search engine through Proteome Discoverer (version 1.4.1.14) (Thermo Scientific). The details of the functional activities of each protein are in the added Supplementary table 2.

On page 6, the authors state: “SDS-PAGE gel of the major fraction showed a strong band at about 75kDa (Supplementary Fig. S3C).”

- How was the 75 KDa band analyzed? Details of this should be included in the Methods section or in the Supplementary materials.

The band was analyzed using a mass spectrophotometer (Proteomics facility) as indicated in the "Proteomic analysis" section. As suggested and for the sake of clarity we added Supplementary table S3 with the results of the analysis, showing the list of proteins of the chromatographic peaks.

On page 7, in the legend of Figure 5:

- The legend of Figure 5 does not explain the difference in experimental conditions for Figure 5G versus Figure 5H. Similarly, in Supplementary Figure S6B, what is the different in terms of the experimental conditions between the upper and lower spectra?

Figures 5G and 5H correspond to the same experimental conditions. We made this point explicit in Figure 5 legend, line 403 ("same experimental conditions").

On page 7, related to Figure 5:

- With respect to the amount of saliva used to treat PE film versus PE 4000 granules, how much PE were used per saliva per treatment was used for the RAMAN, FTIR and HT-GPC analyses? Does the amount of PE:Saliva ratio affect the efficiency of enzymes?

In relation to PE film and PE 4000 in the HT-GPC analysis, we used 20 mg. We added this information in the Methods, section "High Temperature-Gel Permeation Chromatography (HT-GPC) analysis", line 57.

As for the film used in the RAMAN and FTIR experiments, the saliva was applied on the surface of the film. The quantity of PE film affected by the saliva depends on the surface in contact with the saliva, which is difficult to estimate.

The question about the PE:Saliva ratio and the efficiency of the enzymes is a very interesting one. However, in these experiments we aimed at showing the effect of the saliva on PE, a phenomenon not described up to this work. The efficiency of ww saliva and the isolated enzymes, alone or in combination, is a part of a larger and longer-term project. These kinds of experiments are currently very difficult to carry on, and in our opinion at this point they would not change the message contained in the manuscript.

On page 9, in the Discussion: The authors state, "Unexpected instead is the capacity of these particular proteins to oxidize PE, a polymeric, compact hydrophobic substance. However, the ecological niche of Lepidoptera and the potential necessity to react to plant phenols might provide a possible explanation."

- This part of the discussion needs to be revised. It seems that the authors do not know the biology of the Greater Waxmoth, *G. mellonella*. It is no surprise that larvae of *G. mellonella* secrete enzymes that can degrade hydrocarbons because *G. mellonella* larvae live in beehives and consume beeswax. This should be incorporated into the discussion.

We thank the Reviewer for this comment. We modified the Discussion according to the Reviewer suggestion, line 270: "The wax worms live and grow in the honeycombs of the beehives. They feed among other things (pollen, larvae etc.) on beeswax. Given the similarity between plastics and beeswax, it is conceivable that the observed effect on PE is a consequence of the warm capability to degrade wax."

On page 15, the authors state: "Larvae of 150-300 mg were used for saliva collection. Briefly, a glass capillary connected to a mouth pipet was placed at the buccal opening and the liquid was collected."

While the authors provide information about the mass of the larvae used, they do not provide any information about the volume of saliva collected at any one time, or the protein concentration in the saliva samples.

We added the missing information in the Materials and Methods, section “Wax worm saliva collection”, line 26 (“Five to ten microliters of saliva were collected from each worm. The concentration of proteins in the saliva measured via Bradford methodology varies between 20 and 30 mg/ml”).

On page 17, the authors state: “Mass spectra *.raw files were searched against an in-house specific database against Galleria proteins (12715 proteins entries)”.

- What is the nature of the “Galleria protein database”? Was this derived from an annotated genome sequence of *G. mellonella*? Is there a url link to this data base?

The data are available in Data Mendeley ([doi:10.17632/t7b5s58vxt.2](https://doi.org/10.17632/t7b5s58vxt.2)), as specified in the Materials and Methods, section “Data and Code Availability”, line 15.

Suggested improvements: This reviewer suggests running FTIR on the PE treated with Demetra and Ceres separately to compare with the very first FTIR analysis of wax worm saliva on PE film. When Demetra and Ceres are present together, the FTIR spectra shows peaks that represent carbonyl-groups. Demetra by itself can show visual deterioration of PE, while Ceres does not. An additional FTIR analysis would show the generation of the carbonyl-groups from wax worm saliva is due to either Demetra or Ceres. In addition, kinetics of PE degradation by Demetra and/or Ceres with respect to protein quantity (mass) vs time would be very informative.

We provided FTIR analysis of PE treated with the two PEases, confirming a greater oxidative effect of Demetra compared to Ceres, line 224, in Results, section “Identification of wax worm enzymes as PE oxidizers”. We added Supplementary Figure S8.

We agree with the Reviewer that kinetics of PE degradation by the enzymes is an important aspect of this line of research, particularly in view of future application. The results of Figure 6 and Figure S2 show that the effect of Demetra and GmSal, respectively, increases with time, which is a measure of the reaction kinetics, albeit a relative one. Measuring degradation in absolute terms is a complex issue. Using the GC-MS, we would need an exhaustive list of degradation products, even the ones that for technical reasons escape detection, being absolutely sure that we have them all. Then we would need to make a standard curve for each of that and use this curve as an internal standard to quantify the production of each of them. Even then, we would have an indirect estimation of PE degradation. This is a different issue than standard measuring of metabolic activity using CO₂ release. Finding alternative experimental approaches to achieve this kind of quantification might be difficult in the current conditions, as we would need large amount of proteins, something that is not trivial to-date. We strongly believe that this will be an important information, but we think that the message the manuscript delivers does not depend on this particular aspect. This is the first time that animal enzymes capable to degrade PE have been found, an effect that increases with increasing the number of applications/time.

In relation to “a more comprehensive comparison between the two should be done including chemical structure of the two, substrate specificity, sequence comparison and other biochemical characterization” we think that these would be indeed valuable information. The interest of this point resulted evident by the in-silico analysis of the structural and functional differences between the two enzymes that we performed and added to the text (new section in Materials and Methods, entitled “Analyses of the wax worm PEases sequences”, line 198).

We added this paragraph: “Both enzymes present the same functional domains as hemocyanins’ (Figure S9A, B). However, the sequence comparison between the two enzymes shows only 30% of identity (Figure S9 C). Moreover, in silico analysis (Figure S9 D) indicates that Demetra is more stable than Ceres, which could be one of the reasons contributing to the observed differences”, in Results, section “Identification of wax worm enzymes as PE oxidizers” (line 236).

As indicated in the text, differences in stability might account for the variation in the experimental outcomes.

However, we believe that providing additional experimental evidences of all this information is well beyond the scope of a single paper on the matter: the main message of the article is to point out that the saliva of this invertebrate is responsible for the observed PE degradation, together with the identification of two specific enzymes with PEase activity. The chemical structure is an altogether different and very complex project, as is an exhaustive biochemical characterization; the specificity is a very interesting question but at this point it is not clear what the substrate of these enzymes might be, as indicated in the Discussion. We believe that each of these questions opens a new line of research. Simultaneously addressing all these lines is beyond the scope of this manuscript. We think that this discovery will attract the interest of the scientific community, opening up and widening the approach and perspective on this new issue.

Reviewer's expertise: This reviewer has extensive expertise in the microbial and enzymatic degradation of PE and other natural and synthetic polyester polymers.

Reviewer #2 (Remarks to the Author):

The paper from Sanluis-Verdes, Colomer-Vidal, and colleagues investigates the enzymes found in wax worm saliva for polyethylene degradation. Overall, this is an incredibly interesting paper and I strongly endorse this study for urgent publication. This work will be extremely interesting to many in the research community focused on plastics biodegradation.

We thank the Reviewer for his/her very positive comments on the manuscript.

The authors, if room permits, should remove the sentence about “the first report of ...” and use that extra space to instead state what the enzymes are. (?)

We changed the sentence in the abstract following the Reviewer suggestion. It now reads as follows: “Within the saliva, we identified two enzymes, belonging to the phenol oxidase family, that can reproduce the same effect. These enzymes are the first animal enzymes with this capability, which opens the way to new ground-breaking solutions for plastic waste management through bio-recycling/up-cycling”.

The sentence “[i]n fact, no such enzyme has been identified yet, confirming the crucial limiting role of oxidation in the whole biodegradation process chain” is an illogical statement to me. Because an enzyme has not been identified does not necessarily mean that oxidation is the crucial limiting role – this should be rewritten. I will note that I fully agree that oxidation is critical here – this is merely a comment on the logic flow of the sentence.

We definitely agree with the Reviewer comments. We changed the sentence to: “Abiotic pre-treatments such as radiation or heat causes oxidation of the polymer, which is the crucial limiting step in the whole biodegradation chain¹²”, in Introduction, line 77.

Why do the authors name the enzymes Demetra and Ceres? This is seemingly a reference to Greco-Roman mythological characters, but the enzyme names do not convey any useful information. The authors should use a more appropriate and informative name.

We agree with the Reviewer that a more functional definition would be better.

Following the Reviewer advice, we introduced the category of PEase for these enzymes (line 95-Introduction, line 226-Results, and lines 273 and 304-Discussion). However, we would like to keep these two names to differentiate the two enzymes.

We decided to call them Demetra and Ceres because they are the names of the agricultural goddess (in Greek and Roman mythology, respectively), and by extension, Nature. We thought that it would be appropriate for these enzymes.

What does “physical deterioration of PE” mean?

We agree with the Reviewer and removed the ambiguous term “physical”.

Figure 1 – By “picks” in the figure caption, I assume the authors mean “peaks”?

We thank the Reviewer for pointing it out. It was supposed to be “peaks”, we changed it accordingly (line 373).

Maybe the authors could show the actual films themselves as they go through treatment? That would be potentially really nice to see.

We agree with the Reviewer that it would be nice. Actually, we tried to do something like that, but it was technically difficult for the lack of adequate equipment (a scope with proper magnification, a camera and availability for the duration of the experiment).

I am wondering about the experimental protocols. Why was 90 minutes chosen as a reaction time and why was fresh enzyme or ww saliva then added? The choice of 90 minutes was arbitrary, on the base of incipient evaporation of the sample (in the case of enzymes), and the apparent oxidation/aggregation of the saliva content at room temperature. Did the authors run at much longer times with a single inoculum? We did not for the previous reasons. In their experimental setup, did they dilute the ww saliva? No, we did not dilute the saliva.

For the 6-day experiment, did the authors use HT-GPC to measure the MW distributions as well? That could be quite interesting. The number of applications of the 6-day experiment (Fig S2) and the first point of the HT-GPC are the same (15 applications). Therefore, the answer is yes. What is the extent of measurable product release in these experiments? That would be nice to quantify on a C-mole basis from the original substrate if possible.

We agree with the Reviewer that to know the absolute measure of product release would be nice. The results of Figure 6 and Figure S2 show that the effect of Demetra and GmSal, respectively, increases with time, which is a measure of how degradation progresses. As we explained to Reviewer 1, “the measurement of degradation in absolute terms is a complex issue. Using the GC-MS, we would need an exhaustive list of degradation products, even the ones that for technical reasons escape detection, being absolutely sure that we have them all. Then we would need to make a standard curve for each of that and use this curve as an internal standard to quantify the production of each of them. Even then, we would have an indirect estimation of PE degradation. This is a different issue than standard measuring of metabolic activity using CO₂ release.

Finding alternative experimental approaches to achieve this kind of quantification might be difficult in the current conditions, as we would need large amount of proteins, something that is not trivial to-date. We strongly believe that this will be an important information, but we think that the message the manuscript delivers does not depend on this particular aspect. This is the first time that animal enzymes capable to degrade PE have been found, an effect that increases with increasing the number of applications/times. “

I think that the authors would benefit the community if they were to include their full proteomics dataset in the manuscript, or (better yet) upload those data to a web server that is (post-publication) publicly accessible. I suspect that Nature Communications requires that anyway.

We definitely agree with the Reviewer. We added all the proteomics dataset in new Supplementary tables 2 and 3.

Are the enzymes in the arylphorin sub-domains of an intact enzyme, by chance? As far as we know they do not correspond to any sub-domain of other enzymes.

We evaluated whether the gene models of Ceres and Demetra were complete. We observed that a complete open reading frame is present in both the sequences, in addition, a 3'-UTR is annotated in both the gene models. Finally, similar proteins we detected in other insects through BLAST all show a similar length and domain composition. Overall these observations point to the fact that the two gene models should be complete and the protein sequence intact

How did the authors inactivate the enzymes in the referenced control experiment?

The enzymes were inactivated at 100 degrees centigrade for 10 minutes as indicated in the Materials and Methods, section “RAMAN and FTIR analyses”: “For the control with inactivated proteins, recombinant proteins were denatured at 100 degrees for 10 minutes” (line 39).

The lack of protein characterization here is quite deflating. This would make this paper (in my opinion) so much better, and seems like an obvious missed opportunity. At an absolute minimum, the authors should do a BLAST search to look for similar enzymes that have been characterized biochemically and/or structurally to see if these types of putative activities are unprecedented or not. The authors allude to the fact that others will need to follow-up on this discovery – certainly many will do so, but I think the authors could do further service by including at least literature review around these enzymes and closely-related homologues.

We added the BLAST search in new Supplementary tables S4 and S5 with a list of NCBI proteins showing sequence identity above 50%. Moreover, we added Supplementary table S6 indicating which of those proteins have been structurally characterized. We changed the text accordingly (Discussion, line 298). We added a new section in Materials and Methods, entitled “Analyses of the wax worm PEases sequences”, line 198).

We thank the Reviewer for this suggestion, as it provided very useful insights. Particularly, this information points out the fact that the study of this type of enzymes is an incipient field, where all the structural biochemical and molecularly detailed functional descriptions are in need. Demetra and Ceres are the first PEases ever found. As suggested by the Reviewer, we added a description in the Discussion (line 276) in relation to these enzymes and closely related homologues, and related references (reference 54-59, in Reference section).

I think the authors should tone down or remove the “no pretreatment” statement – if animals are chewing the polymers, that’s pretreatment.

The pre-treatment statements refer to normally used abiotic pretreatments (heat, radiation...). In all the described experiments, plastics were never in contact with the animal, so no chewing of the polymers occurred.

Did the authors attempt to express the enzymes in *E. coli*? We did indeed try to express these enzymes in different *E. coli* strains, but unfortunately the attempts were not successful.

REVIEWERS' COMMENTS

Reviewer #1 (Remarks to the Author):

The authors have adequately addressed the comments and concerns of the previous review. Well done. This reviewer feels that the manuscript is now ready for publication.

Reviewer #2 (Remarks to the Author):

I still do not like giving enzymes arbitrary names. The motivation from Nature here is a nice sentiment, but arbitrary names add no value to the scientific literature in my opinion, and for example, Yoshida and coworkers in their 2016 Science paper could have just as easily called the *Ideonella sakaiensis* PETase Demetra, considering it comes from Nature too. Much better would be giving the enzymes functional names that are related to their function.

Reply to REVIEWER COMMENTS

We thank Reviewer 1 for his/her positive evaluation.

Reviewer 2 comment.

I still do not like giving enzymes arbitrary names. The motivation from Nature here is a nice sentiment, but arbitrary names add no value to the scientific literature in my opinion, and for example, Yoshida and coworkers in their 2016 Science paper could have just as easily called the *Ideonella sakaiensis* PETase Demetra, considering it comes from Nature too. Much better would be giving the enzymes functional names that are related to their function.

Reply: Following the Reviewer's comments, we adopted the terminology PEase in the previously submitted version of the manuscript. To differentiate between the two PEases, we have not found any formal definition. The use of "trivial" names is also accepted for enzymes (see <https://www.ebi.ac.uk/intenz/rules.jsp>)

Any choice of names to differentiate between the two enzymes (for instance PEase A and PEase B) would be equally arbitrary. We believe that the two chosen names Demetra and Ceres might be easier to remember than simply PEase A and B, particularly as we have already defined these two enzymes as PEases. We hope that the Reviewer will find this choice acceptable.